# A Rare Case of Precocious Puberty in a Child with a Novel *GATA-4* Gene Mutation: Implications for Disorders of Sex Development (DSD) and Review of the Literature

**DOI:** 10.3390/genes14081631

**Published:** 2023-08-16

**Authors:** Tommaso Aversa, Giovanni Luppino, Domenico Corica, Giorgia Pepe, Mariella Valenzise, Roberto Coco, Alessandra Li Pomi, Malgorzata Wasniewska

**Affiliations:** 1Department of Human Pathology of Adulthood and Childhood, University of Messina, Via Consolare Valeria 1, 98125 Messina, Italy; tommaso.aversa@unime.it (T.A.); giovilup97@gmail.com (G.L.); domenico.corica@unime.it (D.C.); giorgia.pepe@unime.it (G.P.); mvalenzise@unime.it (M.V.); cocoroberto93@gmail.com (R.C.); alessandra.lipomi92@gmail.com (A.L.P.); 2Pediatric Unit, AOU Policlinico G. Martino, Via Consolare Valeria 1, 98125 Messina, Italy

**Keywords:** cryptorchidism, disorder/differences of sex development, *GATA-4* gene, micropenis, precocious puberty

## Abstract

Background: Disorders/Differences of sex development (DSD) are often due to disruptions of the genetic programs that regulate gonad development. The *GATA-4* gene, located on chromosome 8p23.1, encodes GATA-binding protein 4 (GATA-4), a transcription factor that is essential for cardiac and gonadal development and sexual differentiation. Case Description: A child with a history of micropenis and cryptorchidism. At 8 years of age, he came under our observation for an increase in sexual pubic hair (pubarche). The laboratory parameters and the GnRH test suggested a central precocious puberty (CPP). Treatment with GnRH analogs was started, and we decided to perform genetic tests for DSD. The NGS genetic investigation showed a novel and heterozygous variant in the *GATA-4* gene. Discussion: In the literature, 26 cases with 46,XY DSD due to the *GATA4* gene were reported. Conclusion: The novel variant in the *GATA-4* gene of our patient was not previously associated with DSD. This is the first case of a DSD due to a *GATA-4* mutation that develops precocious puberty. Precocious puberty could be associated with DSD and considered a prelude to hypogonadism in some cases.

## 1. Introduction

Disorders/Differences of sex development (DSD) represent a heterogenous group of congenital conditions with abnormal development of internal and/or external genitalia [1]. Anormal programs of embryological development, genetic mutation, and hormone alterations can correlate with these disorders [2].

DSDs are estimated to affect 1:4500 newborns and include different abnormalities of urogenital differentiation divided into several categories, according to the 2006 Chicago Consensus classification: chromosomal DSD, 46,XX DSD, and 46,XY DSD [3].

DSD related to sex chromosomes are the most frequent class of DSD, and they include syndromic conditions such as Turner Syndrome (TS), Klinefelter Syndrome (KS) and mixed gonadal dysgenesis. KS is characterized by the 47,XXY karyotype and its incidence is about 1 in 500 males [3]. During the neonatal period, micropenis with hypospadias and/or cryptorchidism are the most evident sex disorders. During the puberal period, the penis undergoes normal development, but definitive testicular volume develops with delay. In addition, these patients could develop gynecomastia, female-type fat deposition, and fertility invalidated by azoospermia and hypogonadism [2,3].

The 46,XX DSDs are caused by an excess of androgens that lead to precocious virilization. This group presents ovotesticular and testicular cases. Classical congenital adrenal hyperplasia (CAH) is the most common disease of 46,XX DSD, and it is due to a significantly reduced activity of 21-hydroxylase that determines two subtypes: the simple virilizing and the salt-wasting forms. Clinical manifestations of this condition are already present at birth, with ambiguous genitalia and electrolyte alterations appearing after a few weeks when there is no residual enzyme activity (Salt-wasting form) [3,4]. 46,XX DSD includes other cases of adrenal enzyme deficiency and rare conditions such as ovotesticular karyotype 46,XX. Patients with 46,XX DSD present anomalies of genitalia, a disorder of puberal development, and potential infertility [5].

Complete Androgen Insensitivity Syndrome (CAIS) and Persistent Mullerian ducts syndrome (PMDS) belong to the 46,XY DSD family. The incidence of 46,XY DSD can vary considerably, from rare disorders, such as complete gonadal dysgenesis affecting 1/20,000 births, to common conditions, such as hypospadias affecting 1/250 male births [6]. 46,XY DSD includes patients with defects in androgen biosynthesis, impaired testosterone action, and abnormal testicular differentiation with different virilization entities [7]. The complete absence of virilization entails female external genitalia, and the diagnosis could occur in the pubertal age because of a lack of breast development and/or primary amenorrhea. Partial virilization is often noted at birth for abnormal genital anatomy. The latter situation could be characterized by the association of different conditions such as micropenis, cryptorchidism, hypospadias, and incomplete scrotal fusion [8]. Abnormalities of gonadal development and sex hormone defects are among the main causes of micropenis and cryptorchidism. Genetic conditions, such as syndromes or specific mutations, define the copresence of these disorders. *GATA4* haploinsufficiency, a rare cause of 46,XY DSD, could cause a wide spectrum of genital abnormalities, including micropenis, cryptorchidism, and hypospadias, as reported in the 26 patients with DSD due to the *GATA4* mutations that are included in previous studies [9,10,11,12,13,14,15,16,17,18,19].

GATA-binding proteins (GATA), a group of six transcriptional regulator factors, interact with specific nucleotide sequences thanks to zinc finger DNA-binding domains. The *GATA4* gene, located on chromosome 8p23.1, is essential for cardiac and gonadal development. GATA-binding protein 4 (GATA4) regulates the expression of multiple genes’ coding for hormones or components of the steroidogenic pathway during testis development and function. *GATA4* is expressed in the gonads together with *GATA1* and *GATA6*, but only *GATA4* is involved in early gonad development [20]. Regulating the expression of sex-determining region Y (*SRY*), SRY-box transcription factor 9 (*SOX-9*), and anti-Mullerian hormone (*AMH*) genes, *GATA4* initiates testis differentiation [21]. In Sertoli cells, *GATA4* appears as an amplifier of AMH transcription. In Leydig cells, *GATA4* modulates a couple of steroidogenic genes. A blockade of *GATA4* expression in mice Leydig cell lines leads to the suppression of the steroidogenic program and a decrease in hormone production [22].

Within the pituitary gland, GATA transcription factors have been shown to be critical for pituitary cell differentiation and function: GATA2 regulates gonadotropins gene expression and GATA2-GATA4 stimulates basal and GnRH-induced ADCYAP1, a polypeptide able to stimulate LH and FSH secretion by primary pituitary cells and the gonadotrope cell line [23,24].

Even if *GATA4* has an important role in the development of congenital heart disease (CHD), DSD and CHD can be isolated or combined in patients with *GATA4* mutations [15]. Heterozygous *GATA4* mutations have been identified in more than 140 patients with ventricular septal defect, tetralogy of Fallot, or other cardiac malformations [25].

The aim of this study was to present the peculiar case of an 8-year-old child with 46,XY DSD, central precocious puberty (CPP), and a novel *GATA4* mutation without CHD. In addition, we aimed to review the characteristics of DSD cases related to *GATA-4* alterations to understand if additional clinical findings could link the *GATA-4* pathway with precocious puberty.

## 2. Clinical Report

The child is a second-born of healthy non-consanguineous parents, born at 40 gestational weeks by cesarean section after a normal pregnancy. At birth, he had mild respiratory distress: his Apgar scores were 6 and 8 at 1′ and 5′, respectively. His birth weight and length were 3900 gr and 54 cm, respectively. The family history was negative for endocrinological pathologies. Genitalia examination at birth showed a micropenis (length 1.5 cm) and scrota without gonads palpable. Due to the presence of micropenis (1.5 × 1.0 cm), in the second month of life, a treatment of intramuscular testosterone at a dose of 25 mg/month was administered for 3 months. When testosterone therapy was started, the patient had a length of 58.3 cm (+2.1 SDS) and bilateral undescended testis. Then, the patient was lost at follow-up, and when he was 2 years old, in a different hospital, orchidopexy was performed for bilateral cryptorchidism. After 6 years, when he was 8 years old, the patient was referred again to our endocrinology clinic for a precocious pubarche. Height was 147.1 cm (+2.95 SDS), weight was 42.0 kg (+2.21 SDS), and bone age was significantly advanced (11 years and 3 months versus 8 years of chronological age). Physical examination showed hypertelorism, a singular 5 cm wide cafe-au-lait spot in the abdomen, curly pubic hair, testes palpable in the scrota with a volume of 3 mL, and a penis length of 4 cm. For suspected precocious puberty, appropriate investigations were performed during hospitalization, such as the GnRH test, and are reported in Table 1. At the GnRH test, the LH peak was 74.7 mIU/mL with an elevation of IGF1, testosterone, LH, and FSH basal levels. At ultrasonography, testicular volumes were reported at 1 mL bilaterally. Considering the patient’s age, male gender, and medical history, the suspicion of intracranial neoplasia was excluded by the normal pituitary magnetic resonance imaging (MRI). The echocardiography and thyroid ultrasonography were normal. Considering a diagnosis of central precocious puberty, a treatment with GnRH analogs was started, even if the elevation of gonadotrophins after the GnRH test could suggest a future diagnosis of hypergonadotropic hypogonadism. Psychological counseling had not demonstrated behavioral or personality changes. In addition and considering that the history of microphallus and bilateral cryptorchidism together with the high testosterone basal levels, in the absence of befitting testicular volumes, could be an expression of gonadal dysgenesis, it was decided to perform genetic investigations for DSD. Chromosome analysis revealed a 46,XY karyotype. An array-CGH study identified a de novo 815 Kb micro-deletion in the 2q34 region of the *ERBB4* gene, previously never described in association with DSD and precocious puberty. A Next Generation Sequencing (NGS) panel of genes related to gonadal dysgenesis and precocious puberty was adopted. The gene panel included *AMH*, *AMHR2*, *AR*, *CBX2*, *CYPIIAI*, *CYP11AI*, *DHH*, *DMRTI*, *GATA4*, *HSDI7B3*, *MAP3KI*, *NROBI*, *NR5A1*, *SOX9*, *SRD5A2*, *SRY*, *WNT4*, *WTI*, *WWOX*, *KISS1*, *KISSIR*, and *MKRN3.* The NGS analysis identified a heterozygous and missense variant c.671CG (p.Ser224Cys) in the *GATA-4* gene.

Segregation analysis in the parents is still being carried out. Furthermore, genetic investigations have ruled out other conditions such as McCune–Albright syndrome.

## 3. Discussion

The genetic causes of 46,XY DSD are determined in almost half of the cases, although the latest genetic techniques (such as whole genome sequencing, WGS; and whole exome sequencing, WES) can identify new genes and novel mutations [5]. In a large cohort study with 682 DSD cases, there were 119 cases of 46,XY DSD. Only 71 of the total patients with 46,XY DSD underwent WES. In the XY cohort, the main genetic causes are pathogenetic variants in the *AR* and *NR5A1* genes. A 3-year-old patient assigned at birth as a female, with typical female genitalia and dwarfism, was the only 46,XY DSD case with the *GATA4* mutation (p.G12R) [19]. In another large international cohort study, the *AR* and *NR5A1* genes were seen more frequently in 46,XY DSD patients and only 1–2% of cases were related to the *GATA4* gene [11,13]. *GATA4* haploinsufficiency is therefore a rare cause of DSD in genetic males.

In the literature, 26 cases with 46,XY DSD due to the *GATA4* gene have been reported until now (Table 2) [9,10,11,12,13,14,15,16,17,18,19]. Only five cases were raised as female. The main clinical phenotype reported was unilateral or bilateral cryptorchidism (17 out of 26 cases), and it was associated with micropenis, hypospadias, and other clinical elements of DSD. Complete gonadal dysgenesis and female external genitalia were reported in three patients with different genotypes. Furthermore, the most frequent *GATA4* variants were p.P407Q (31%) and p.G221R (12%). The P.407Q variant was found in a female 46,XY with complete female genitalia, in four males with cryptorchidism and micropenis, and in three males with hypospadias and cryptorchidism. In experimental studies [13,15] this variant causes decreased expressions of the *AMH* and *SRY* genes. Instead, two patients presented the same variant, p.W228C, and similar clinical phenotype characterized micropenis and cryptorchidism [11,14]. According to Van den Bergen et al. [12], the p.W228C variant is situated on the N-terminal zinc finger and the loss of interaction with ZFPM2 protein revealed its pathogenicity and the alteration of gonadal development. Anomalies of the N-terminal zinc finger alter the binding and activation of some of the *GATA4* interaction partners. In fact, p.P226L and p.T228C variants act on the N-terminal region without changing the overall structure of GATA4 [14].

CHD was found in five patients, and three cases of them had atrial septal defects. Yui Shichiri et al. suggested that *GATA4* activity is mostly deficient in patients with both cardiac defects and DSD than that observed in patients with only CHD [18]. This conclusion was derived by analyzing the expressivity of the p.Pro163Ser variant of *GATA4* in five different patients with CHD of which only one also presented a DSD. In this way, they assumed that normal development of the heart required stricter regulation of *GATA4* transcription than gonadal development [18].

Different variants of the *GATA4* gene are responsible for 46,XY DSD but the phenotypes, ranging from a mild under-virilization to complete female external genitalia, are different in patients with the same mutation, and it could be linked to a variable expressivity and penetrance of the *GATA4* gene [17]. Analyzing manifestations of patients, cryptorchidism, hypospadias and micropenis are the clinical phenotypes most related to the *GATA* gene. In fact, clinical presentations of our case were micropenis ad cryptorchidism without a family history of DSD. NGS investigation of our patient, for DSD panel genes, showed heterozygous and missense variant, c.671CG, in the *GATA-4* gene. This variant was classified as likely pathogenic according to The American College of Medical Genetics and Genomics (ACMG), and it causes punctiform mutation, p.Ser224Cys, in the specific aminoacidic sequence. The C.671CG variant was not previously associated with DSD, described in scientific literature, nor noted in reference databases. A particular additional finding in our case was the precocious pubarche that led to a diagnosis of precocious puberty. Other peculiar elements were the elevated basal levels of testosterone together with a significant elevation of LH and FSH, which could suggest a hypergonadotropic hypogonadism. None of the 26 cases with 46XY DSD due to the *GATA4* gene already reported presented precocious puberty, while some cases have been documented as physiological pubertal development. Wagner-Mahler K et al. described a follow-up of over almost 15 years of a child born with perineal hypospadias, micropenis, cryptorchidism, and 8p23 deletion [16]. Subsequently, he developed slight psychomotor-behavioral impairment and microcephaly without facial dysmorphic features or neuroimaging alterations. At the age of 11, the child started spontaneous puberty with asymmetric development of the testicles (better testis development in the left compared to the right), and at the age of 13.5 years, replacement therapy with testosterone heptylate was started due to no progression in sexual development. After puberty, the patient developed a left ventricular noncompaction. Lourenço et al. identified a heterozygous *GATA4* missense mutation (p.G221R) in three 46,XY DSD patients from one family [9]. The index case had only DSD, and his spontaneous puberty began at 12 years and was complete by 14 years. His brother had micropenis and bilateral cryptorchidism associated with atrial septal defect and a pubertal development beginning at 12.5 years. They proposed that the DSD observed in 46,XY carriers of the *GATA4* mutation (p.G221R) was due to a loss-of-function effect on *GATA4* in the developing gonad, interfering with its ability to interact with *FOG2*, and disturbing activation of the *AMH* promoter together with *NR5A1* [9]. This conclusion was supported by Stefan White et al. [10], to understand the mechanism that linked 8p23 deletion and clinical phenotype (complete gonadal dysgenesis, female external genitalia, and congenital adrenal hypoplasia) in a male 46,XY DSD.

We analyzed the literature to find cases of precocious puberty in patients with 46,XY DSD caused by monogenic alterations. This research allows us to identify two cases. Eggers et al. [11] analyzed DNA from the largest reported international cohort of patients with DSD (278 patients with 46,XY DSD and 48 with 46,XX DSD). Only one case with *LHCGR* mutation (2p16.3) had presented precocious puberty with 46,XY DSD, altered gonadal differentiation, and Leydig cell hypoplasia. Lele Li et al. [26] evaluated the clinical manifestations and genetic variants of 36 cases with 46,XY DSD due to the *MAMLD1* variant. Hypospadias was the most prevalent phenotype among 10 cases, but it was not present in a male patient with normal external genitalia at birth. The latter case developed premature secondary sexual characteristics (increased penis size, growing beard, and pubic hair) at the age of 1 year and in addition, he showed elevated LH, FSH, and T-levels which indicated precocious puberty. His testosterone level later decreased, and so he displayed hypergonadotropic hypogonadism. The patient’s genotype was a missense variant of *MAMLD1*, p.A421P, classified as VUS according to ACMG criteria [26]. Like *GATA4*, the *MAMLD1* gene plays a molecular role in the pituitary gland level and its mutations could be a rare cause of DSD.

Moreover, CPP could be a clinical element in patients with DSD. In fact, some patients with chromosomal DSD also presented precocious puberty. Chunxiu Gong et al. [27] reported a rare case of Klinefelter syndrome (KS) with CPP and performed a review of the literature that showed different cases in which precocious sex development was found in KS, Turner Syndrome, and congenital adrenal hypoplasia. Some KS cases had abnormal productions of human chorionic gonadotropin via endocrine tumors as pathogenic elements of peripherical precocious puberty. However, the pathogenetic mechanisms of CPP in patients with Klinefelter syndrome are not entirely understood. It is assumed that early activation of the HPG axis may result in CPP and subsequent hypogonadism, due to premature gonadal cell apoptosis, in patients with KS. GnRH agonist therapy allows the suppression of the HPG axis to treat precocious puberty and protect against premature gonadal insufficiency [27].

## 4. Conclusions

The *GATA4* mutation is a rare cause of 46,XY DSD with variable expressivity and genetic penetrance that can lead to different phenotypes. We encountered precocious puberty as a new clinical element in a patient with 46,XY DSD and testicular dysgenesis, probably secondary to a novel *GATA4* mutation. The combination of CPP and DSD manifested in our patient with a pathologic baseline increase in FSH and LH, secondary testosterone, and the clinical signs of virilization, accelerated growth, and bone maturation. Precocious puberty could be considered an additional finding in the clinical phenotype of DSD patients, although pathophysiological processes have yet to be understood. To the best of our knowledge, the novel missense variant of our case, c.671CG, was not already described, and this is the first patient in which precocious puberty is associated with a DSD due to a *GATA4* mutation.

## Figures and Tables

**Table 1 genes-14-01631-t001:** Patient’s laboratory data at the presentation of precocious pubarche.

	Laboratory Data	Reference Values *
Basal FSH (mIU/mL)	71.30	0.26–3
Peak FSH (mIU/mL)	178	6.7–24.5
Basal LH (mIU/mL)	11.5	0.02–5
Peak LH (mIU/mL)	74.7	<5.0
Total Testosterone (ng/dL)	300	<10
Estradiol (pg/mL)	<5	0.3–1
Delta-4_Androstenedione (ng/mL)	0.62	<0.5
ACTH (pg/mL)	36.40	11.0–82.0
Cortisol (ug/dL) at 8 a.m.	10.80	5.0–25.0
DHEAS (ug/dL)	51.10	42–109
17-OH Progesterone (ng/mL)	0.50	0.1–2.7
IGF-1 (ng/mL)	633	68–255
α fetoprotein (ng/mL)	7.58	<10

FSH: follicular stimulating hormone; LH: luteinizing hormone, ACTH: adrenocorticotropic hormone, DHEAS: dehydroepiandrostenedione sulphate; IGF-1: insulin-like growth factor 1. * Reference values according to age and prepubertal stage.

**Table 2 genes-14-01631-t002:** Overview of 46,XY DSD cases due to GATA-4 mutations.

References	Cases	Genotype	DSD Manifestation	CHD	Other Manifestations
Lourenço D. et al., 2011 [9]	1	p.G221R	Male: Perineal hypospadias, hypoplasia of corpus cavernosum, fused hypoplastic labioscrotal fold, bilateral cryptorchidism.		
	2	p.G221R	Male: bilateral cryptorchidism and microphallus	ASD	
	3	p.G221R	Male: Fused labioscrotal folds, hypospadias, bilateral cryptorchidism		
White S. et al., 2011 [10]	4	8p23 deletion	Male: Complete gonadal dysgenesis, female external genitalia		AHC
Eggers S. et al., 2016 [11] Van den Bergen J.A. et al., 2020 [12]	5	p.W228C	Male: Micropenis and cryptorchidism		
	6	p.A346V	Male: perineal hypospadias, chordee, cryptorchidism, penoscrotal transposition.		
	7	p.P394T	Male: Perineal Hypospadia.		
	8	p.P394T	Female: no virilization, no uterus, inguinal bilateral testes		
	9	p.P407Q	Male: Cryptorchidism and penile hypospadias		Imperforate anus
	10	p.P407Q	Male: Scrotal hypospadias, hypoplastic uterus, testes palpable.		
	11	p.P407Q	Male: Perineal hypospadias and cryptorchidism		
Igarashi M. et al., 2018 [13]	12–16	p.R265C (n.1) and p.P407Q (n: 4)	5 Males: hypospadias with or without micropenis and cryptorchidism		
Martinez de LaPiscina I. et al., 2018 [14]	17	p.C238R	Female: Clitoral hypertrophy, fused labia with posterior raphe, gonads palpable in inguinal canal, rudimentary uterus.	VSD	Autism
	18	p.W228C	Male: micropenis, hypospadias and bilateral cryptorchidism		
	19	p.P226L	Male: micropenis and bilateral cryptorchidism		Obesity
Choi JH et al., 2019 [15]	20	p.R215G	Male: micropenis, bilateral cryptorchidism, perineal hypospadias		
	21	p.P407Q	Female: complete famale genitalia		
Wagner-Mahler K. et al., 2019 [16]	22	8p23 delation	Male: Perineal hypospadias, bilateral cryptorchidism, bifid scrotum, Mullerian structures absent, micropenis.	LVN	
Nurullah Çelik et al., 2021 [17]	23	p.T113P	Male: microphallus, scrotal hypoplasia, bilateral cryptorchidism	ASD, VSD, PS	Umbilical hernia
	24	p.P163S	Male: microphallus, bifid scrotum and perineoscrotal hypospadias		Ptosis
Yui Shichiri et al., 2022 [18]	25	c.C487T	Female: typical female external genitalia.	ASD	Hiatal hernia, CH, epilepsy
Evgenia Globa et al., 2022 [19]	26	p.G12R	Female: typical female genitalia		Dwarfism
Our patient	27	c.671CG	Male: micropenis and cryptorchidism		Central Precocious Puberty

AHC: adrenal hypoplasia congenita, ASD: atrial septal defect, CH: congenital hypothyroidism, LVN: left ventricular noncompaction, PS: pulmonary stenosis, VSD: ventricular septal defect.

## Data Availability

Data are available from the authors upon request.

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
