# Peer review of "A Rare Case of Precocious Puberty in a Child with a Novel GATA-4 Gene Mutation: Implications for Disorders of Sex Development (DSD) and Review of the Literature"

_genes, 2023, doi:10.3390/genes14081631_

Round 1

Reviewer 1 Report

The manuscript “Central Precocious Puberty in a Child with 46,XY DSD and Novel GATA4 Mutation: Case Report and Review of the literature” is an interesting Case Report.

Unfortunately, the actual manuscript presents too many formal deficiencies to be able to be considered for publication.

Authors should address the following aspects:

- 1) English language throughout the manuscript. Review language and orthograph.

- 2) Gene abbreviations in italic letters.

- 3) Disorders/Differences of sex development (DSD) are the terms now accepted.

- 4) Abstract, line 22: I would indicate a “central precocious puberty (CPP).”

- 5) Introduction, line 34: “… abnormal development of internal and/or external genitalia…”

- 6) Introduction: authors try to describe the different types of DSD, although the description is chaotic. Among the 46,XX DSD, there are ovotesticular and testicular cases.

- 7) Lines 71-74: the references here cited do not follow the order !

- 8) Lines 136-137: add as Supplementary Material the list of candidate genes for gonadal dysgenesis and precocious pubertal included in the panel.

- 9) Table 1: give reference values for the patient age and not for adult males.

- 10) Table 2: for patient 4, indicate as “Other manifestations”: AHC (Adrenal Hypoplasia Congenita) and not CAH (this the usual abbreviation for “congenital adrenal hyperplasia”).

- 11) Line 269: I would not say that “… precocious puberty is linked to DSD and GATA4 gene mutation”…. This is rather an association of a CPP in a boy with a 46,XY DSD and a testicular dysgenesis probably secondary to a GATA4 mutation. CPP can be independent of the DSD cause and the combination manifests here with a pathologic baseline increase of FSH and LH, and secondary of T, and the clinical signs of virilization, accelerated growth and bone maturation.

The manuscript “Central Precocious Puberty in a Child with 46,XY DSD and Novel GATA4 Mutation: Case Report and Review of the literature” is an interesting Case Report.

Unfortunately, the actual manuscript presents too many formal deficiencies to be able to be considered for publication.

Authors should address the following aspects:

- 1) English language throughout the manuscript. Review language and orthograph.

- 2) Gene abbreviations in italic letters.

- 3) Disorders/Differences of sex development (DSD) are the terms now accepted.

- 4) Abstract, line 22: I would indicate a “central precocious puberty (CPP).”

- 5) Introduction, line 34: “… abnormal development of internal and/or external genitalia…”

- 6) Introduction: authors try to describe the different types of DSD, although the description is chaotic. Among the 46,XX DSD, there are ovotesticular and testicular cases.

- 7) Lines 71-74: the references here cited do not follow the order !

- 8) Lines 136-137: add as Supplementary Material the list of candidate genes for gonadal dysgenesis and precocious pubertal included in the panel.

- 9) Table 1: give reference values for the patient age and not for adult males.

- 10) Table 2: for patient 4, indicate as “Other manifestations”: AHC (Adrenal Hypoplasia Congenita) and not CAH (this the usual abbreviation for “congenital adrenal hyperplasia”).

- 11) Line 269: I would not say that “… precocious puberty is linked to DSD and GATA4 gene mutation”…. This is rather an association of a CPP in a boy with a 46,XY DSD and a testicular dysgenesis probably secondary to a GATA4 mutation. CPP can be independent of the DSD cause and the combination manifests here with a pathologic baseline increase of FSH and LH, and secondary of T, and the clinical signs of virilization, accelerated growth and bone maturation.

Reviewer 2 Report

The article "A Rare Case of Precocious Puberty in a Child with a Novel GATA-4 Gene Mutation: Implications for Disorders of Sex Development (DSD)" provides a comprehensive and intriguing account of a unique clinical case involving a child with micropenis and cryptorchidism. The study delves into the genetic basis of the patient's condition by examining the GATA-4 gene, which is crucial for gonad development, sexual differentiation, and cardiac development.

The implications of this study are limited due to the single case presentation. Studies with larger cohorts or systematic investigations are generally more impactful in rare disorders like DSD.

In Leydig cells, GATA4 modulates a couple of steroidogenic genes. But in this case, the testosterone level is normal. Can the authors explain that?

Round 2

Reviewer 1 Report

The 2nd version of the manuscript genes-2543215, has been improved.

There are still several formal corrections to be applied:

- 1) Line 98: GATA4 in italic.

- 2) Lines 135-137: gene abbreviations in italic. Moreover, there are several mistakes in gene abbreviations, look at: CYP11A1, NR5A1, SRD5A2

- 3) Along the text: if gene abbreviations should have italic letters, the mutations (c….. or p. ….) are in normal letters: look at lines 138 , 153, 162, 163, 167, 168, 171, 172, 183, 194 (also gene not in italic),196,197, 212, 217, 236

- 4) In Table 2:

            - patient 4: 8p23 deletion and the abbreviation AHC

            - patient 9: imperforate

            - patients 12-16: hypospadias

- 5) Line 180: “CHD was ….”

- 6) Line 182: “This conclusion ….”

- 7) Line 246: “….. understood.”

English language has been improved. Still several corrections to be applied.

Reviewer 2 Report

The authors have well addressed the comments and I recommended it to be accepted. 

Author Response

The authors are very grateful to this Reviewer for the revision and for the accept our paper. Thanks.